# A new synthetic medium for the optimization of docosahexaenoic acid production in *Crypthecodinium cohnii*

**Pingping Song**[1]*, **Alexander Kuryatov**[2], **Paul H. Axelsen**[2]

**1** School of Biological Engineering, Guizhou Medical University, Guiyang, Guizhou Province, China,
**2** Departments of Pharmacology, Biochemistry and Biophysics, and Medicine, University of Pennsylvania, Philadelphia, Pennsylvania, United States of America

☯ These authors contributed equally to this work.
* songqian70@163.com

**Data Availability Statement:** All relevant data are within the paper.

**Funding:** This work was supported by National Institutes of Health AG057197 from Dr Paul H.

## Abstract

The heterotrophic microalgae *Crypthecodinium cohnii* was usually cultivated in complex medium containing glucose, yeast extract and sea salt. For the preparation of DHA with highest purity, a new defined medium without the yeast extract was developed. Different inoculated densities, C/N ratios, temperatures, culture volumes and glucose additions were investigated to optimize the algal growth rate and DHA production. The growth period in *C. cohnii* was shortened from 12–14 days to 7–8 days, the $OD_{600}$ was enhanced from 2.0 to 3.0, the glucose consumption was accelerated and used up on day 3–4, and the DHA content in culture were increased from 10 to 45 nmoles/300 μl batch. It was found that *C. cohnii* had optimal growth and DHA accumulation in 25 ˚C, 0.2 inoculated density, 5–10 C/N ratio, 5:1 air/culture volume ratio. This is the first time DHA production using *C.cohnii* has been optimized in synthetic medium. This allows preparation of uniformly radiolabeled $^{13}$C- and $^{14}$C-DHA.

## 1. Introduction

In recent years, Docosahexaenoic acid (DHA) belonging to the ω-3 polyunsaturated fatty acid (PUFA) has become the focus of considerable attention due to its neuroprotective effects [1,2]. DHA, enhanced in phospholipids in the central nervous system and retina, is recognized to multiple roles in brain health and diseases, such as Alzheimer's disease [3–5].

The traditional source of ω-3 fatty acids is fish oil, which contains two major PUFAs, DHA and eicosapentaenoic acid (EPA) [6]. However, the fish oil as a food additive has significant problems: its strong fishy smell, unpleasant taste and poor oxidative stability [7]. Marine micro-organisms that contain large amounts of DHA might provide this important fatty acid [8,9]. Microalgae biomass is exceptionally suitable for the extraction and isolation of individual PUFAs due to its stable and reliable contents [10]. Hence, screening of microalgae for DHA production is significant.

Axelsen; Guizhou Science and Technology Foundation of China (No. TS [2017]-2843, TJ [2017] 1143, GY [2017] 5-6) from Dr Pingping Song.

**Competing interests:** The authors have declared that no competing interests exist.

*Crypthecodinium cohnii* (*C. cohnii*) has been recognized as a high-volume producer of DHA amongst the marine dinoflagellate microalgae, especially in pharmaceutical and nutraceutical applications. This microalgae is exceptional in that it can accumulate a high fraction of DHA with trivial amounts of other PUFAs in cell lipids, which makes the DHA isolation very interesting, particularly for higher incorporation of glucose into DHA [11,12]. DHA-producing *C. cohnii* is cultivated in media containing the glucose as a sole carbon source, so the media compositions are very important to the DHA accumulation in *C. cohnii*.

DHA production from *C. cohnii* has been extensively studied and optimized [7]. High-level manufacturing of algal fatty acids is feasible by using traditional fermentation systems. Only a limited number of studies of purity of the DHA extracted from *C. cohnii* have been reported. In animal experiments, carbon isotope radiolabeled polyunsaturated fatty acids with high purity are usually used to explore the pathogenesis of some diseases [13]. However, the media used to grow *C. cohnii* are not very useful for preparing DHA samples with high isotopic purity. Here we reported a better synthetic defined medium for growing *C. cohnii*. The effects of various medium components and culture conditions on algal growth and DHA production were studied.

## 2. Materials and methods

### 2.1 Materials

DHA was purchased from Nu-Check Prep Inc. (Elysian, MN). d5-DHA was purchased from Cayman Chemicals (Ann Arbor, Michigan). *C. cohnii* (ATCC 40750) was obtained from American Type Culture Collection (Manassas, VA, USA). All other chemicals were obtained from Sigma-Aldrich (St. Louis, MO, USA).

### 2.2 Medium and culture conditions

*C. cohnii* stocks were cultivated in 4 g/L yeast extract, 12 g/L glucose, 35 g/L sea salt medium at 26°C in the dark. New cultures were inoculated to an $OD_{600}$ of about 0.15. After 4–5 days $OD_{600}$ reached ~1.5, and 1.5–2 ml of this culture were centrifuged at 500 g for 1 min. The supernatants were discarded and the left pellets were washed in ~5 ml of defined media (without glucose and glutamate) and centrifuged at 500 g for 1 min. After this, pellets were re-suspended in 10 ml of synthetic medium at different glutamate concentrations (1.8 g/L, 0.9 g/L, 0.6 g/L, 0.45 g/L, 0 g/L), temperatures (15°C, 20°C, 25°C, 30°C), culture volumes (5 ml, 10 ml, 15 ml, 20 ml) and additional glucoses concentrations (9 g/L, 4.5 g/L, 1.5 g/L, 0.5 g/L). The final $OD_{600}$ were adjusted to 0.15–0.2, and cultures were incubated in a refrigerated incubator shaker (Edison, NJ, USA) in the dark at 200 rpm. The growth and DHA content of *C. cohnii* were analyzed daily.

A defined medium developed by Tuttle and Loeblich [14], were shown in following Tables 1, 2 and 3. All stock solutions were sterilized by the filtration through 0.22 μm Milex syringe filters. Variations on this defined medium were investigated in the experiments which follow.

### 2.3 The growth rate and glucose determination

The growth rate was determined by measuring the $OD_{600}$ using a Cary 400 Bio UV-Vis spectrophotometer (Agilent, Santa Clara, CA). Glucose consumption was measured by the DNS method [15,16]. After the centrifugation of algal culture, the supernatant (25 μl) were taken and added to 275 μl water, then 300 μl DNS (1 g/L 3,5-dinitrosalicylic acid, 0.1 g/L $Na_2SO_3$, 1 g/L NaOH) were added, and incubated for 10 min at 90°C. After the incubation, 600 μl quencher (40 g/L sodium potassium tartrate) were added, and the final solution was cooled

**Table 1. The defined medium compositions per liter.**

| Reagents | Contents |
|---|---|
| glucose | 9 g |
| $K_2HPO_4$ | 1 g |
| $MgCl_2.6H_2O$ | 10.6g |
| $CaCl_2$ | 1.1 g |
| KCl | 0.7 g |
| $Na_2SO_4$ | 3.9 g |
| $SrCl_2.6H_2O$ | 0.1 g |
| KBr | 0.1 g |
| NaCl | 23.5 g |
| $NaHCO_3$ | 0.2 g |
| Disodium glycerophosphate | 0.15 g |
| glutamate | 1 g |
| metal mixture | 5 ml |
| vitamin solution | 1 ml |

The pH was adjusted to 6.4.

**Table 2. The metal mixture compositions in defined medium.**

| Reagents | Concentrations (g/L) |
|---|---|
| $FeCl_3.6H_2O$ | 0.5 |
| $Na_2EDTA$ | 10 |
| $H_3BO_3$ | 10 |
| $CoCl_2.6H_2O$ | 0.01 |
| $MnCl_2.4H_2O$ | 1.6 |
| $ZnCl_2$ | 0.1 |

down to room temperature. Glucose concentration was measured by $OD_{540}$ and analyzed by the glucose standard curve.

## 2.4 Lipid extraction and saponification

300 μl of algal culture in a 1.5 ml Eppendorf tube were centrifuged for 1 min at 2000 g. The supernatant was removed, and the pellet was re-suspended in 640 μl water. The suspension was subjected to three free-thaw cycles (liquid nitrogen alternating with boiling water), and cooled down on ice. 1.6 ml methanol and 800 μl dichloromethane were added and mixed, then

**Table 3. The vitamin mixture compositions in defined medium.**

| Reagents | Concentrations (mg/L) |
|---|---|
| Thiamin | 100 |
| Vitamin $B_{12}$ | 5 |
| Aminobenzoate | 20 |
| Ca pantothenate | 10 |
| Biotin | 3 |
| Riboflavin | 100 |

sonicated 90 s on ice. 800 µl dichloromethane and 640 µl water were added to separate phases, which were centrifuged at 400 g for 1 min. The lower phase was withdrawn and transferred to new 13x100 mm glass tubes, and dried under argon.

Pellets were dissolved in 1.5 ml of 85% methanol in water, then 0.5ml of 1 M NaOH was added. Saponification was performed at 80°C for 1 h. Then the tubes were cooled to room temperature. The samples were acidified with 400 µl HCl (5 M). Then 1 ml of isooctane were added, vortexed and allowed to settle before the upper phase was pipetted into a glass tube. The combined upper phases were evaporated under argon. 100 µl of ethanol were added, transferred to the new vials with cap, blown with argon and then the vials were stored in a -80°C freezer.

## 2.5 HPLC separation and mass spectrometry analysis

An aliquot of calibrated amount of d5-DHA were added to the extracts of DHA to calculate the content. 5 µl samples were injected into a 1.0x50 mm Eclipse XD8-C18 3.5 µm column. Solvent A was 60% acetonitrile, 40% water and 0.1% formic acid. Solvent B was 100% acetonitrile and 0.1% formic acid. The mobile phase was pumped at 0.1 ml/min. From 0 to 5 min the solvent was 0% buffer B, then the composition was changed linearly from 0–100% solvent B at 5–6.5 min, 100% solvent B at 6.5–10 min, returned to 0% at 10–10.5 min, and then kept at 0% buffer B from 10.5 to 12.5 min. The eluent was alkalinized post-column with 0.15 M ammonium hydroxide in methanol flowing at 50 µl/min, and introduced into an ABI 4000 Q1 Trap tandem mass spectrometer (Sciex, Toronto, Canada) via electrospray ionization at negative polarity. The declustering potential (DP) was -70 V, the ionspray voltage (Is) was -4000 V, the temperature of drying gas (TEM) was 200°C, the collision energy (CE) was -20 V and the collision gas (CAD) was 4psi for multiple reaction monitoring (MRM) mode. The m/z transitions in MRM mode were from 327.2–283.2 for $^{12}C$-DHA with the neutral loss of $CO_2$, 332.2–288.2 for d5-DHA.

## 3. Results

### 3.1 Growth and DHA production of *C. cohnii* under different conditions

**3.1.1 Different inoculated densities.** When the inoculated density was 0.4, the algal cells had the highest growth rate, the maximum $OD_{600}$ was 2.5, but the highest DHA content (0.1 µmole/batch) in culture was got on day 4 when the inoculated density was 0.2 (Fig 1A and 1B).

**3.1.2 Different glutamate concentrations.** In the N-deprived group, algal cells didn't grow, the glucose was consumed very slowly, and DHA content in culture was lowest during the 7-day culture period. From the 0.45 g/L to 1.8 g/L of glutamate concentration, growth rate, glucose consumption and DHA production of *C. cohnii* gradually increased. At 1.8g/L glutamate concentration, the growth rate was the highest, and reached a maximum $OD_{600}$ on day 5, then kept at the stable level on days 6–7 (Fig 2A). From 0.9 g/L to 1.8 g/L of glutamate concentration, the glucose consumption in culture was rapid over days 1–4, then stopped in 2 g/L on days 5–7 (Fig 2B). DHA production was the highest on day 4, then declined sharply (Fig 2C).

**3.1.3 Temperatures.** At 15°C, algal cells grew slowly, glucose was not consumed efficiently, and DHA production was very low during the incubation (Fig 3A–3C). At 20°C, algal cells grew faster, and reached a maximum $OD_{600}$ on day 6 (Fig 3A), but the glucose consumption stopped on day 7 (Fig 3B). DHA production was highest on day 6 (Fig 3C). At 25°C, the growth rate of *C. cohnii* was the highest (Fig 3A). Glucose consumption was most rapid over days 1–3, then stopped at 2 g/L on days 4–7 (Fig 3B). DHA production kept at a higher level

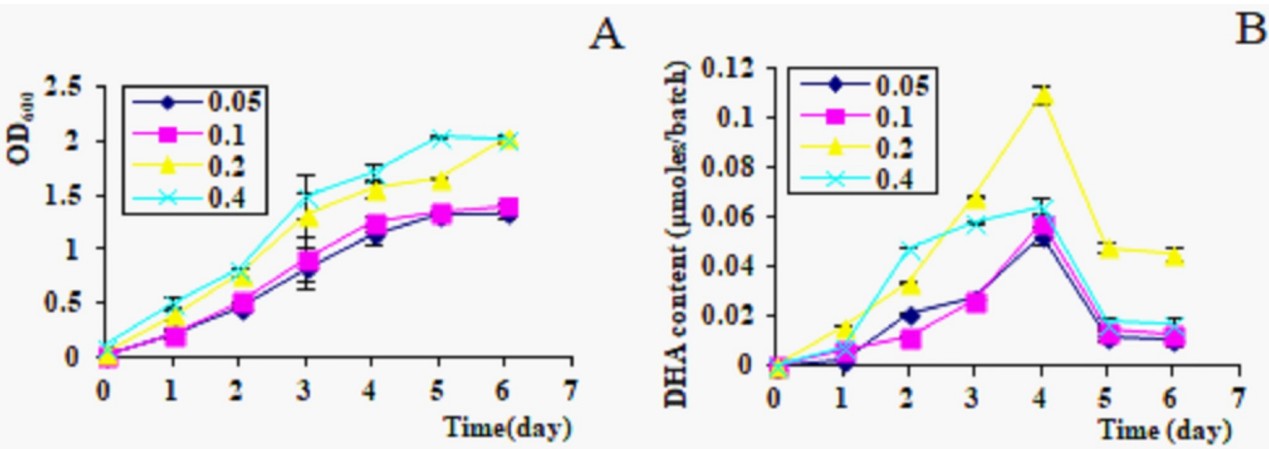

**Fig 1. Growth rate and DHA production in different inoculated densities.** A. $OD_{600}$; B, DHA content in culture, 1ml/batch.

(Fig 3C). At 30°C, algal cells grew fast at first, then declined sharply after day 5 (Fig 3A). Glucose consumption stopped at 2 g/L after day 5, and DHA production was low (Fig 3B and 3C).

**3.1.4 Air/Culture volume ratios.** Growth rate of algal cells was highest in 9:1 and 4:1 air/culture volume ratios. Glucose consumption stopped on days 3–7. DHA production reached the highest level on day 4. In 1.5:1 air/culture volume ratio, both growth rate and DHA yield were the lowest, and glucose consumption stopped on days 5–7 (Fig 4A–4C).

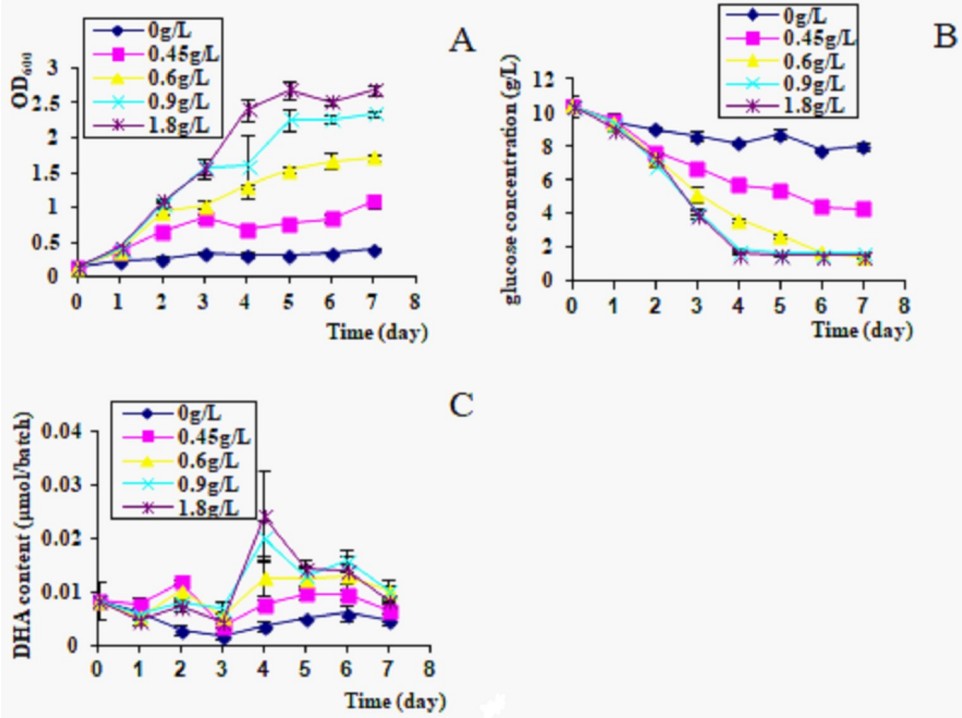

**Fig 2. Growth rate and DHA production in different glutamate concentrations.** A, $OD_{600}$; B, glucose consumption; C, DHA content in culture, 300 μl/batch.

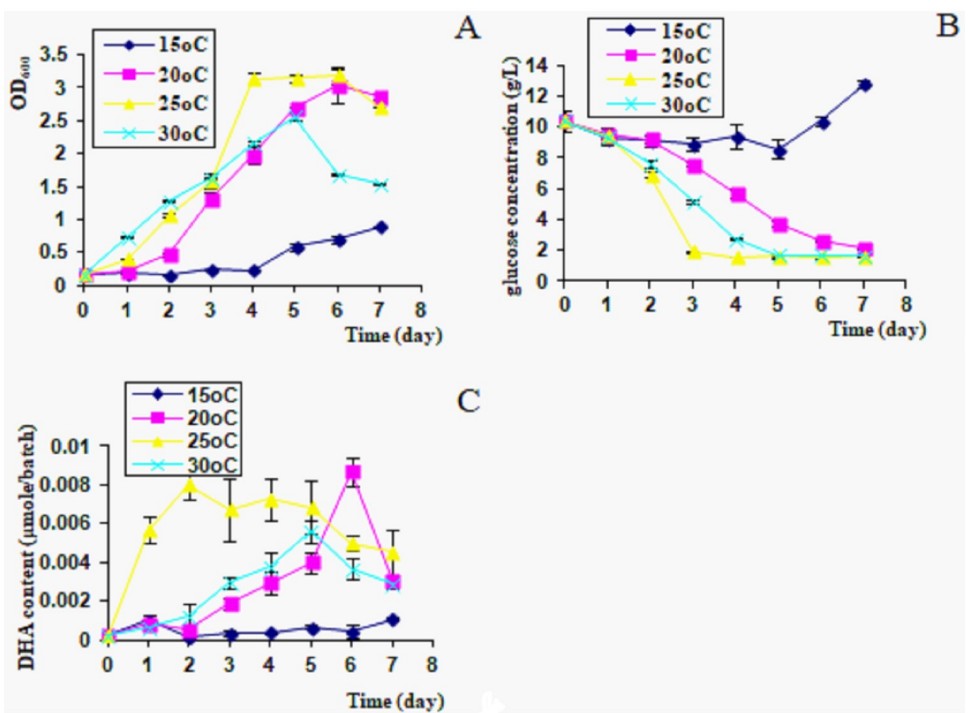

**Fig 3. Growth rate and DHA production in different temperatures.** A, $OD_{600}$; B, glucose consumption; C, DHA content in culture, 300 μl/batch.

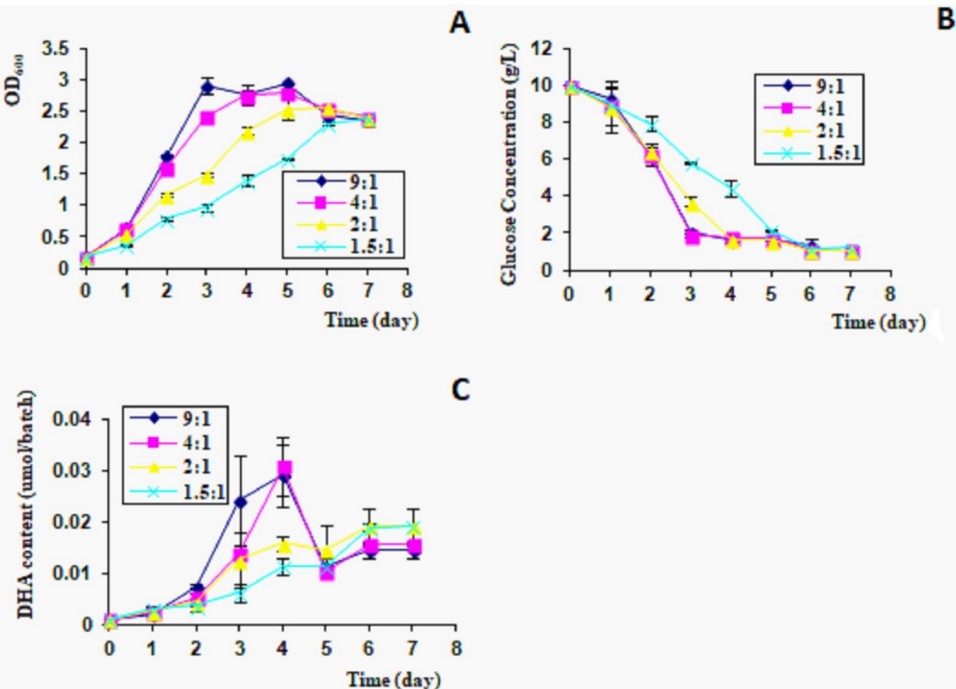

**Fig 4. Growth rate and DHA production at different air/culture ratios.** 5 ml (9:1), 10 ml (4:1), 15 ml (2:1), and 20 ml (1.5:1) of medium in 50 ml tubes. A, $OD_{600}$; B, glucose consumption; C, DHA content in culture, 300 μl/batch.

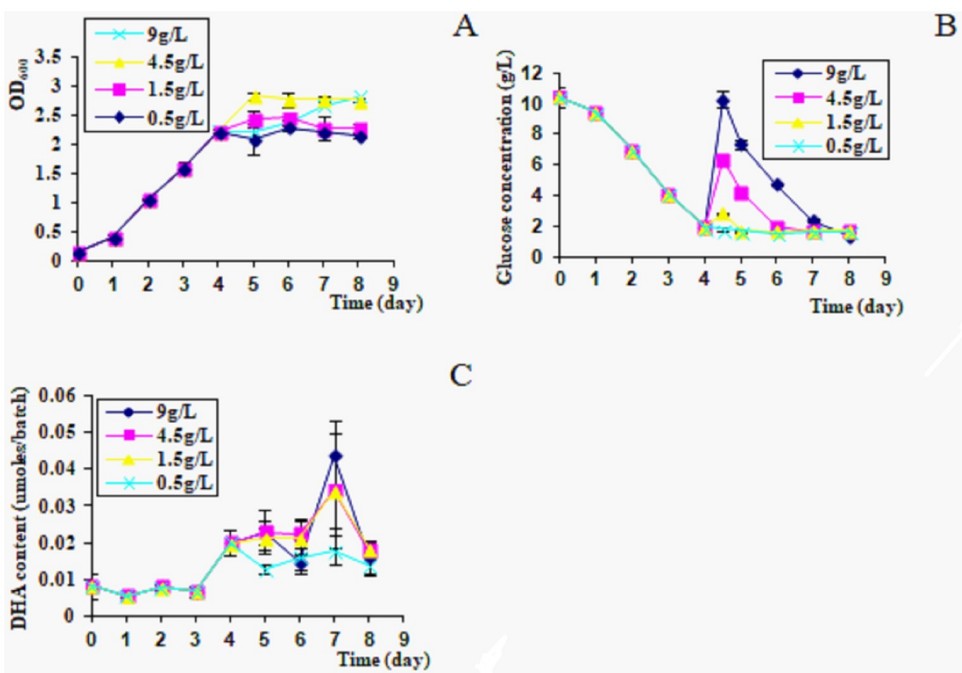

**Fig 5. Growth rates and DHA productions with adding additional glucose.** A, $OD_{600}$; B, glucose consumption; C, DHA content in culture, 300 μl/batch.

**3.1.5 Adding supplementary glucose.** DHA content of *C. cohnii* in culture was 0.02 μmoles/batch on day 4. After adding 9 g/L glucose, DHA accumulation reached 0.045 μmoles/batch on day 7. After adding 4.5 g/L glucose, *C. cohnii* had a maximum $OD_{600}$ on day 5, DHA content in culture was 0.035 μmoles/batch, which was similar with 1.5 g/L glucose on day 7. After adding 0.5 g/L glucose, $OD_{600}$ and DHA accumulation were both lower than other groups (Fig 5A–5C).

## 4. Discussion

In these experiments, the growth rate and DHA production from *C. cohnii* in different conditions were analyzed.

Many kinds of complex media with different components have been investigated to culture the *C. cohnii* [17–19]. All of them used yeast extract. In our studies an optimal defined medium was found. In this medium, a metal mixture and various vitamins were added to enrich the culture. $OD_{600}$ was used to analyze the algal growth. The growth period of algal cells was significantly shortened from 13–14 days to 7–8 days (data not shown). Algal cells generally entered the growth phase on day 2, reached the highest growth rate ($OD_{600}$ = 2.5–3) on days 4–5, and reached the plateau phase on days 5–7.

Different inoculated densities from *C. cohnii* were investigated in these experiments. When the original inoculated density was about 0.2, algal cells achieved the optimal growth rate and highest DHA production (Fig 1B). Higher or lower inoculated densities were not beneficial for algal growth and DHA accumulation [20]. Considering uniformly [13]C or [14]C-labeled DHA experiments, the ideal inoculated density of *C. cohnii* was determined to 0.15–0.2.

C/N ratio plays a very important role for algal growth and DHA production [21,22]. Carbon and nitrogen are both necessary nutrient elements in the medium. When nitrogen was efficiently consumed in culture, glucose was transformed into more lipids. In our experiments,

it was also found that the growth and DHA production of *C. cohnii* had differed depending on different glutamate concentrations (Fig 2A and 2C). Algal cells didn't grow in N-deprived condition, glucose was consumed very slowly, and DHA content was the lowest during the culture period. Growth rate and DHA production of *C. cohnii* gradually increased when glutamate concentrations were raised from 0–1.8 g/L. At 0.9–1.8 g/L glutamate, glucose consumption was most rapid, so the algal cells achieved a higher $OD_{600}$ and DHA yield. The optimal C/N ratio for algal growth in *C. cohnii* was determined to be 5–10.

Temperature is also an important factor for algal growth and DHA accumulation. At a low temperature (15˚C), *C. cohnii* had a slow growth rate and low DHA production (Fig 3A and 3C). At a high temperature (30˚C), algal growth and glucose consumption were restricted (Fig 3A and 3B). Even though the DHA production was relatively higher on day 7, the cultivation period was prolonged. 25˚C was found to be the optimal temperature for growth and DHA production (Fig 3A and 3C). Algal cells grown at different temperatures may adapt by changing the proportion of polyunsaturated fatty acids in order to maintain proper membrane lipid fluidity and cell functions [23,24].

Volume ratio of air and culture was a key factor for algal growth [25]. Some reports have indicated excess culture in limited volume decreased the contact area with air and thus reduced the oxygen supply and ventilation efficiency [26]. *C. cohnii* was cultivated in 50 ml tube. For 5 ml and 10 ml cultures, the algal cells had higher growth rate, glucose consumption stopped on day 3, and the DHA production reached the highest level on day 4 (Fig 4A–4C). But in 20 ml cultures, the growth rate and DHA production of *C. cohnii* were the lowest (Fig 4A and 4C), and glucose consumption stopped two days later (Fig 4B). So, the optimal volume ratio between air and culture was 4–9.

After addition of glucose, *C. cohnii* had a higher growth rate, but DHA accumulation was not increased (Fig 5a and 5C). After adding extra 9 g/L glucose, DHA content in culture on day 7 was two-fold higher than on day 4, but was much lower than without glucose addition (Fig 5C). Hence, it was concluded that glucose addition can't promote DHA accumulation in *C. cohnii*. Although glucose could be transformed into lipids during metabolism, excessive glucose could accelerate the algal growth and consumption of cellular lipids.

By these optimizations, optimal condition for the growth rate and DHA production in *C. cohnii* were established. These findings would provide optimal methods for the biosynthesis of uniformly radiolabeled [13]C- and [14]C-DHA, and for the research on the fate of oxidation products of DHA in Alzheimer's disease.

## Acknowledgments

The authors would like to thank Prof. Jon Lindstrom (University of Pennsylvania, Philadelphia) for proofreading and corrections of the manuscript, Dr Hiroaki Komatsu and Dr Ran Furman (University of Pennsylvania, Philadelphia) for their help with experimental technologies, and Dr Chris Moser (University of Pennsylvania, Philadelphia) for his assistance with the incubator shaker.

## Author Contributions

**Conceptualization:** Paul H. Axelsen.

**Formal analysis:** Pingping Song.

**Funding acquisition:** Paul H. Axelsen.

**Investigation:** Pingping Song, Alexander Kuryatov.

**Project administration:** Paul H. Axelsen.

**Resources:** Paul H. Axelsen.

**Supervision:** Paul H. Axelsen.

**Writing – original draft:** Pingping Song.

**Writing – review & editing:** Alexander Kuryatov.

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
