## [Decision Letter · Decision Letter 0]

11 Feb 2020

A New Synthetic Medium for the Optimization of Docosahexaenoic Acid Production in Crypthecodinium cohnii

PONE-D-19-32556

Dear Dr. Song,

We are pleased to inform you that your manuscript has been judged scientifically suitable for publication and will be formally accepted for publication once it complies with all outstanding technical requirements.

With kind regards,

Prakash Kumar Sarangi, PhD

Academic Editor

PLOS ONE

Reviewers' comments:

Reviewer's Responses to Questions

**Comments to the Author**

1. Is the manuscript technically sound, and do the data support the conclusions?

Reviewer #1: Yes

Reviewer #2: Yes

2. Has the statistical analysis been performed appropriately and rigorously? 

Reviewer #1: No

Reviewer #2: Yes

3. Have the authors made all data underlying the findings in their manuscript fully available?

Reviewer #1: Yes

Reviewer #2: Yes

4. Is the manuscript presented in an intelligible fashion and written in standard English?

Reviewer #1: Yes

Reviewer #2: Yes

5. Review Comments to the Author

Reviewer #1: The article is written well and supported by appropriate data. The culture media for C. cohnii is usually a complex media hence a define media for the same is a new development for further research. Though few more attempts using different permutation and combination of minerals and vitamins along with appropriate statistical analyses may be incorporated. The medium developed may be used in scalling up and its performance may be presented.

Reviewer #2: DHA is a very important theme of research and researchers have planned the production of DHA from a renewable source. Cost effective strategies are important and have been taken into consideration. Various parameters affecting growth and production are well discussed.

6. PLOS authors have the option to publish the peer review history of their article (what does this mean?). If published, this will include your full peer review and any attached files.

Reviewer #1: No

Reviewer #2: Yes: Dr. Latika Bhatia

---

## [Editor Report · Acceptance letter]

14 Feb 2020

PONE-D-19-32556 

A New Synthetic Medium for the Optimization of Docosahexaenoic Acid Production in *Crypthecodinium cohnii*

Dear Dr. Song:

I am pleased to inform you that your manuscript has been deemed suitable for publication in PLOS ONE. Congratulations! Your manuscript is now with our production department. 

With kind regards,

on behalf of

Dr. Prakash Kumar Sarangi 

Academic Editor

PLOS ONE